# The Phenotypic Variation in Moso Bamboo and the Selection of Key Traits

**DOI:** 10.3390/plants13121625

**Published:** 2024-06-12

**Authors:** Shihui Zheng, Songpo Wei, Jiarui Li, Jingsheng Wang, Ziyun Deng, Rui Gu, Shaohui Fan, Guanglu Liu

**Affiliations:** 1International Centre for Bamboo and Rattan, Key Laboratory of National Forestry and Grassland Administration, Beijing 100102, China; zsh13472212805@163.com (S.Z.); weisongpo@icbr.ac.cn (S.W.); li1999jr@163.com (J.L.); 15737310333@163.com (J.W.); deng_ziyun@163.com (Z.D.); gr514551314@163.com (R.G.); fansh@icbr.ac.cn (S.F.); 2Yunnan Diannan Bamboo Forest Ecosystem Research Station, Cangyuan County, Lincang 677400, China

**Keywords:** moso bamboo, phenotypic characters, genetic diversity, comprehensive evaluation

## Abstract

This research aimed to explore the diverse phenotypic characteristics of moso bamboo in China and pinpoint essential characteristics of moso bamboo. In this study, 63 grids were selected using the grid method to investigate 28 phenotypic traits of moso bamboo across the entire distribution area of China. The results suggest that the phenotypic traits of moso bamboo exhibit rich diversity, with coefficients of variation ranging from 5.87% to 36.57%. The phenotypic traits of moso bamboo showed varying degrees of correlation. A principal component analysis was used to identify seven main phenotypic trait indicators: diameter at breast height (DBH), leaf area (LA), leaf weight (LW), branch-to-leaf ratio (BLr), leaf moisture content (Lmc), wall-to-cavity ratio (WCr), and node length at breast height (LN), which accounted for 81.64% of the total information. A random forest model was used, which gave good results to validate the results. The average combined phenotypic trait value (D-value) of most germplasm was 0.563. The highest D-value was found in Wuyi 1 moso in Fujian (0.803), while the lowest D-value was observed in Pingle 2 moso in Guangxi (0.317). The clustering analysis of phenotypic traits classified China’s moso bamboo germplasm into four groups. Group I had the highest D-value and is an important candidate germplasm for excellent germplasm screening.

## 1. Introduction

Moso bamboo (*Phyllostachys edulis*) is a member of the Phyllostachys genus of the Gramineae family and is mainly found in subtropical regions. The Ninth National Forest Resources Inventory Report reveals that China’s abundant moso bamboo forest spans 4.67 million hectares, making up 72.96% of the nation’s overall bamboo forest expanse [1]. Phenotypic traits are easily observed, as are measured characteristics produced by the adaptation of plants to different environments during the evolutionary process [2]. The study of phenotypic trait characteristics can reveal the degree of phenotypic variation in plants [3], divide plant germplasm taxa, screen representative traits [4], and is an important basis for the selection of excellent seed sources.

The growth characteristics of moso bamboo resources exhibit notable variations in different geographical areas due to artificial introduction and natural variation over time [5]. For instance, a study identified distinct geographical variability among 17 natural moso bamboo populations in Jiangxi Province, leading to the selection of several dominant moso bamboo germplasm based on four utilization indicators: diameter at breast height, bamboo wall thickness at breast height, height under branches, and culm length [6]. Moreover, the phenotypic characteristics of moso bamboo seeds displayed significant variation across different seed sources, with culm nodes and crown length playing a pivotal role in seed yield [7]. Notably, researchers found that the nutrient characteristics of moso bamboo fine roots varied considerably at different altitudes in the Wuyi Mountain area, suggesting adaptive strategies to the environment [8]. Additionally, a study investigating moso bamboo in six sites within China’s range revealed varying leaf traits across geographical regions [9]. Furthermore, the growth and phenotypic traits of moso bamboo populations in different latitudinal regions (Anhui, Guangxi, and Zhejiang) showed significant differences, with the average length nodes under branches and leaf traits emerging as the most dominant phenotypic traits based on a principal component analysis [10]. Overall, these findings underscore the presence of geographic variation in different moso bamboo populations.

The moso bamboo is widely distributed in 14 provinces across China, from the Qinling Mountains and the Han River Basin to the southern part of the Yangtze River Basin. However, current research on the phenotypic traits of moso bamboo is primarily focused on specific provinces and regions or areas where the distribution of moso bamboo is concentrated. This has led to a relatively narrow focus on phenotypic traits. Therefore, there is a need to expand research on the phenotypic diversity of moso bamboo across the entire range of China. In this study, we used a grid method to systematically examine the phenotypic traits of moso bamboo, aiming to uncover the phenotypic diversity of moso bamboo germplasm in China. Our goal is to explore the relationship between different phenotypic traits of moso bamboo, identify the most important ones, and provide a theoretical basis for selecting excellent moso bamboo germplasm.

## 2. Materials and Methods

### 2.1. Bamboo Germplasm for Testing

The distribution of moso bamboo in China was determined based on the literature, China Digital Herbarium, and a field survey, and 63 grids of 150 km × 150 km were established. Based on the distribution of moso bamboo, two sample points were selected in each grid to investigate the phenotypic characteristics of moso bamboo. Three 20 m × 20 m sample plots were established for each sample point (the interval between sample plots was more than 50 m), and a total of 338 sample plots were established. The survey was conducted to investigate the diameter at breast height (DBH), age, and management history of the moso bamboo in the sample plots. According to the average diameter at breast height of the moso bamboo in the sample plot, five standard plants were selected to study the growth characteristics of the moso bamboo, and one second-degree moso bamboo (2–3 years old) was felled for further determination of culm characteristics, leaf blade morphology, and other traits.

### 2.2. Measurement of Phenotypic Traits

#### 2.2.1. Measurement of Phenotypic Character

Phenotypic traits were determined concerning the Specification for the Description of Forest Germplasm Resources [11] and growth traits (plant height, diameter at breast height, plant crown, branch-to-leaf ratio, water content, and biomass), culm shape traits (diameter at ground level, taper grade, number of nodes of the whole culm, length of node at breast diameter, height under the branch, number of nodes under the branch, thickness at base of pole, thickness at breast height, and wall-to-cavity ratio) and leaf traits (leaf area, leaf shape, leaf dry matter content, leaf thickness, and specific leaf area) were measured in a total of 28 traits.

We used a 0.1 cm diameter tape to measure diameter at breast height, diameter at ground level, and section length at breast height. The SENSSUM EP170 portable electronic scale was used to measure biomass. We used vernier calipers with a precision of 0.01 mm to measure the bamboo wall thickness of the diameter at breast height and the wall thickness at the base of the culm in the four directions of east, south, west, and north to obtain the mean value; we randomly selected 60 leaves from the upper, middle, and lower parts of the culm, and 10 leaves were used as a group to measure the leaf thickness with vernier calipers (0.01 mm) and leaf thickness with a camera (0.01 mm). Leaf thickness was measured with calipers (0.01 mm), photographs were taken with a camera, and leaf length, width, and area were calculated with Image J (2.3.0/1.54d). The leaves, some culms, and branches were taken back to the laboratory, and the weights were determined with an electronic balance with an accuracy of 0.01 g. The leaves were dried in an oven at 105 °C for 30 min; then, the temperature was adjusted to 60 °C to dry to a constant mass, and the corresponding dry mass was measured. The calculation of astringency was carried out using the absolute astringency calculation method.
Wall-to-cavity ratio WCr (mm/mm) = 2 × thorax wall thickness/cavity diameter × 100%
Knot-to-leaf ratio BLr (g/g) = Knot fresh weight/leaf fresh weight × 100%
Specific leaf area SLA (cm^2^/kg) = Leaf area/leaf dry weight

#### 2.2.2. Statistical Analysis

The maximum, minimum, mean, standard deviation, and coefficient of variation in each trait were statistically calculated using Excel 2019 [12]. R 4.3.2 was used to perform a cluster analysis [13], principal component analysis [14], and correlation heat map; random forest modeling of phenotypic trait data was achieved using the R extension package random Forest [15,16]; ArcGis 10.7 was used to draw distribution maps; and the affiliation function was used to generate a comprehensive index score D to evaluate moso bamboo germplasm resources [17,18].

The value of the affiliation function is as follows:(1)$$\begin{array}{c} \mu \;(X_{i})=(X_{i}-X_{min})/(X_{max}-X_{min}),\;i=1,2,\cdots ,n. \end{array}$$
where *X_i_* is the *i*th composite indicator, *X_min_* is the minimum value of the *i*th composite indicator, and *X_max_* is the maximum value of the *i*th composite indicator.

The composite indicator weights are as follows:(2)$$W_{i}=\frac{P_{i}}{\sum P_{i}},i=1,2,\cdots ,n.$$
where *W_i_* is the weight of the *i*th composite indicator among all composite indicators and *P_i_* represents the contribution of the *i*th principal component factor.

The composite indicator superiority is as follows:(3)$$D_{j}=\sum \left[\mu \left(X_{i}\right)\times W_{i}\right],j=1,2,\cdots ,n.$$
where *n* is the number of samples and *D* is the composite indicator assessment value.

## 3. Results

### 3.1. Phenotypic Traits

The coefficients of variation in the phenotypic traits of moso bamboo ranged from 5.87% to 36.57% (Table 1), indicating that the numerical traits of moso bamboo were rich in variation. The coefficient of variation for branch moisture content was the highest at 36.57%, and the coefficient of variation for leaf length to width was the lowest at 5.87%. The degree of variation in traits related to the weight and water content of moso bamboo were larger, both exceeding 20%, indicating that the variation in biomass of moso bamboo was larger, and the morphological indices of moso bamboo were rich in variation in growth, with a high diversity of phenotypic traits; the coefficients of variation in leaf blade traits ranged from 5.87% to 14.69%; and the range of variation in height under branches was from 4.36 to 10.87 m, with a coefficient of variation of 18.56%, which was the most obvious variation in the traits of moso bamboo culms. The coefficients of variation for whole culm node number and breast diameter node length were 7.51% and 7.53%, respectively, which were smaller.

### 3.2. Characterisation of Correlations between Phenotypic Traits

The results of the correlation analysis indicate varying degrees of associations among phenotypic traits (Figure 1). Apart from length of node at breast diameter (LN), branch moisture content, and leaf traits, diameter at breast height (DBH) exhibits significant or highly significant correlations with other traits. The correlation coefficients between DBH and diameter at ground, weight of branches, total weight, and cavity diameter exceed 0.9, while negative correlations are observed with wall-to-cavity ratio and Bmc, with coefficients of −0.50 and −0.11, respectively. Taper grade is highly negatively correlated with LN, leaf thickness, leaf area, leaf width, and leaf length–width ratio. LN shows highly significant positive correlations with leaf thickness, leaf area, leaf length, and leaf width and highly significant negative correlations with branch moisture content, leaf moisture content, leaf length–width ratio, and specific leaf area. Leaf thickness, leaf area, leaf length, and leaf width are highly negatively correlated with culm moisture content and leaf moisture content but are unrelated to branch moisture content. Specific leaf area is significantly positively correlated with branch moisture content, branch moisture content, and leaf moisture content, while exhibiting a highly significant negative correlation with under-branch height; plant crown is highly positively correlated with leaf length–width ratio.

### 3.3. Screening for Key Phenotypic Characteristics

Since there are varying degrees of correlation between phenotypic traits, direct evaluation of the germplasm based on this information will affect its authenticity. The use of a principal component analysis for comprehensive evaluation of the participating germplasm can explain the variation in moso bamboo phenotypic traits with fewer traits. Using an eigenvalue greater than 1.0 as the basis for principal component screening, the first eight principal components were extracted with a cumulative contribution rate of 81.64%, which can better summarize most of the information of the 28 phenotypic traits of the participating germplasm (Table 2). The eigenvalues of the principal components were 9.241, 3.985, 2.335, 1.967, 1.83, 1.29, 1.185, and 1.026, respectively, among which the contribution rate of the first principal component was 33.002%, and the eigenvectors of breast diameter, total weight of moso bamboo, ground diameter, and cavity diameter were larger; the contribution rate of the second principal component was 14.231%, and the eigenvectors of leaf area, leaf width, leaf length, node length at breast diameter, and specific leaf area were larger leaf area; the contribution rate of the third principal component was 8.338%, the eigenvectors of leaf weight, branch and leaf weight, height under branch, leaf length, and leaf aspect ratio were larger. The first and third principal components reflected the biomass of moso bamboo. The contribution rate of the fourth principal component was 7.026%, and the eigenvectors of branch and leaf ratio and sharpness were larger, which reflected the plant type of moso bamboo; the contribution of the fifth principal component was 6.536%, the eigenvectors of leaf water content and specific leaf area were larger, and the second and fifth principal components reflected the phenotypic characteristics of moso bamboo leaf blades; the contribution rate of the sixth principal component was 4.608%, the eigenvectors of wall to cavity ratio, wall thickness at breast diameter and culm water content were larger, reflecting the situation of the morphological characteristics of the wall of the culm of moso bamboo. The largest eigenvectors of the seventh and eighth principal components were branch-to-leaf ratio and node length at breast diameter, respectively. Seven phenotypic traits, namely breast diameter, leaf area, leaf weight, branch-to-leaf ratio, leaf water content, wall-to-cavity ratio, and node length, were extracted from the 28 traits, which were the main factors leading to the differences in phenotypic traits of moso bamboo and were used as important indices for evaluating the germplasm resources of moso bamboo.

### 3.4. Comprehensive Evaluation of Phenotypic Characteristics

By standardizing the 28 trait values of the moso bamboo germplasm and substituting them into the above eight principal components, the eight principal component scores of each germplasm were obtained, the eight principal component scores were normalized using the fuzzy affiliation function method, and the weight coefficients of the eight principal components were calculated (0.404, 0.174, 0.102, 0.086, 0.08, 0.056, 0.052, and 0.045), and then, the composite scores (D-value) of each type of germplasm were calculated (Table 3), and all the germplasm were comprehensively evaluated by D-value. The results showed that the average composite score (D-value) of phenotypic characteristics of moso bamboo germplasm was 0.563, with the highest D-value of Wuyi 1 moso bamboo in Fujian Province (0.803) and the lowest D-value of Pingle 2 moso bamboo in the Guangxi Zhuang Autonomous Region (0.317), indicating that Wuyi 1 moso bamboo had the best comprehensive characteristics and Pingle 2 moso bamboo had the worst comprehensive characteristics.

Correlation analyses were conducted based on 28 phenotypic traits and D-values (Table 4). The results showed that the D-value was positively correlated with 13 trait indices, including diameter at breast height, diameter at ground level, plant height, and number of whole culm nodes, and the correlation reached a highly significant level (*p* < 0.001), was negatively correlated with the wall-to-cavity ratio and reached a highly significant level (*p* < 0.001), whereas diameter at node length, branch-to-leaf ratio, culm water content, leaf water content, leaf thickness, leaf area, leaf length, leaf width, and the composite value of D were not correlated with the composite value of D.

### 3.5. Comprehensive Evaluation of Phenotypic Traits

Based on the above 28 characters, when the Euclidean distance was 62, all the moso bamboo germplasm could be divided into four clusters using the full maximum distance method (Figure 2a), and there were differences in moso bamboo phenotypic traits among the clusters (Table 5). The clustering of germplasm from different provinces was not strictly based on geographical location (Figure 2b).

Group I includes 54 germplasm with a large diameter at breast height, large diameter at ground level, large total biomass, high plant height, high under branching, thick culm wall, high number of nodes in the whole culm, large leaf blade area, small specific leaf area, and low water content in the culm and branches. Overall, the germplasm is excellent, containing all the germplasm from Yunnan, more than 70% of the germplasm from Anhui and Fujian, and more than 38% of the germplasm from Hubei, Sichuan, Chongqing, Zhejiang, Jiangxi, and Hunan.

Group II includes 26 germplasm with a small diameter at breast height, small diameter at ground level, low sharpness, small biomass, low plant height, low height under branches, small branch-to-leaf ratio, and thin culm walls, but long nodes at breast height. The germplasm was poor overall, containing all the germplasm from Henan and 75% of the germplasm from Jiangsu.

Group III includes 25 germplasm with a larger diameter at breast height, large diameter at ground level, large biomass, large sharpness, short node length at breast height, high culm water content, thin leaf thickness, and large leaf aspect ratio and contains all the germplasm from Guizhou and 56% of the germplasm from Jiangxi.

Group IV includes eight germplasm with a large branch-to-leaf ratio, high branch water content, high leaf water content, small leaf area, and larger than leaf area, and contains 50% of the germplasm from Guangxi.

### 3.6. Identification of Different Taxa of Moso Bamboo Germplasm

A random forest discriminant model was constructed to classify and predict the four taxa using seven phenotypic characters, namely, breast diameter, leaf area, leaf weight, branch-to-leaf ratio, leaf water content, wall-to-cavity ratio, and node length. From 113 germplasm, 70% of the samples were selected as the training set, and 30% were made as the independent test set. The random forest algorithm was used to train the training set to construct the prediction model, and the number in the random forest was 500, and when the number of model node variables was 6, the mean of the model misclassification rate was the lowest at 26.58%, and the prediction accuracy of its cross-validation was 70.59% (Table 6). The importance of the phenotypic traits of moso bamboo based on the random forest classification output is shown in Figure 3. The significance of the average reduction in accuracy is in the following order: diameter of breast > leaf water content > leaf area > diameter of breast node length > leaf weight > branch-to-leaf ratio > wall cavity ratio. The established random forest discriminant model can effectively discriminate different types of germplasm and verifies that these seven phenotypic traits can be used as indicators to evaluate the germplasm resources of moso bamboo.

## 4. Discussion

### 4.1. Phenotypic Diversity of Moso Bamboo Germplasm Resources

Plant germplasm resources have been selected naturally and artificially to form the diversity of plant phenotypic traits, and the study of plant phenotypic trait diversity is the basis for the effective organization, conservation, and use of crop improvement [19]. In this study, the mean coefficient of variation in phenotypic traits of moso bamboo was 16.65%. The variation ranged from 5.87% (leaf aspect ratio) to 36.57% (branch moisture content). The coefficients of variation in the traits varied, which were lower than those of *Dendrocalamus lactiferous Munro* (30.84%), *Salix psammophila* (22.53%), and *Ziziphus jujuba* var. *spinosa* (Bunge) Hu ex H.F.Chow. (19. 80%) [20,21,22], indicating that the moso bamboo germplasm has smaller variations than other species, which is similar to previous research [10], but is different in terms of variation in traits. The results of previous studies have shown that in the coefficient of variation in traits, the following order is observed: Thoracic diameter < Thoracic node length < Thoracic wall thickness, whereas with the results of the present study, the following order is observed: Thoracic node length < Thoracic diameter < Thoracic wall thickness, which may be due to the difference in the study area and scale of this study. Among leaf traits, there is a close correlation between specific leaf area and biomass allocation, leaf morphology, and phenotypic plasticity in physiology [23], and the coefficient of variation in specific leaf area of moso bamboo was the largest in leaf morphological traits in this study (14.69%), suggesting that adaptive changes in functional traits of leaf blades of moso bamboo in China are an important strategy to adapt to different growth environments.

### 4.2. Key Phenotypic Traits of Moso Bamboo Germplasm Resources

There were obvious correlations among the traits in this study that reached significant or highly significant levels, and the correlation coefficients between breast diameter, ground diameter, and total biomass exceeded 0.9, which was consistent with the results of studies on conifers [24] and horsetail pine [25]. The correlation coefficient between sharpness and breast pitch length was −0.83, having the largest absolute value of the negative correlation coefficient. Phenotypic traits are not independent of one another [26]; there are varying degrees of correlation among the phenotypic traits of bamboo. Evaluating germplasm solely based on this information may compromise its authenticity. Therefore, we conducted a principal component analysis on all phenotypic traits to identify which traits best represent the overall variation in the data [27]. In this study, seven phenotypic traits, namely, breast diameter, leaf area, leaf weight, branch–leaf ratio, leaf water content, wall-to-cavity ratio, and node length, were screened as the main phenotypic indices in evaluating the germplasm resources of moso bamboo, with cumulative contribution rates of 81% and 64%; at the same time, the seven main phenotypic indices were validated by applying the Random Forest Discriminant Model, which could better reflect the characteristics of moso bamboo germplasm resources from different regions. They can be taken as the key for the next round of research on the phenotypic characteristics of moso bamboo germplasm resources [28].

### 4.3. Evaluation of Moso Bamboo Germplasm Resources

Comprehensive evaluation of moso bamboo germplasm resources through the combination of the affiliation function method and principal component analysis has high reliability and feasibility and has been widely used in studies such as ginkgo [29] and soybean [30]. In this study, the average comprehensive value (D-value) of phenotypic traits of China’s moso bamboo germplasm was 0.563, with the highest D-value (0.803) and the best comprehensive traits for Wuyi 1 moso bamboo germplasm in Fujian Province and the lowest D-value (0.317) and the worst comprehensive traits for Pingle 2 moso bamboo germplasm in the Guangxi Zhuang Autonomous Region. By clustering the phenotypic traits, China’s moso bamboo was divided into four groups, and the evaluation of the D-value of the four groups could also better respond to the status of different types of germplasm, with Group I having the highest D-value, optimal in diameter at breast height, biomass, and wall thickness and greater application value. The lowest D-value was found in Group II, and the growth of moso bamboo germplasm was poor. The range of species distribution area is determined by climatic conditions [31]; in the same climatic environment, moso bamboo obtains similar hydrothermal conditions, and phenotypic traits are similar [32]; the distribution of taxon II in the higher latitude area belongs to the edge of the distribution of moso bamboo germplasm; the hydrothermal conditions are poorer, resulting in the lowest value of D of taxa from Group II, which is similar to the results of cluster analysis by Liu Jiping [33] on the ten climatic factors of the key bamboo-producing areas of China’s moso bamboo. This is similar to the results of Liu Jiping’s cluster analysis of ten climatic factors in key bamboo-producing areas in China. The quality of moso bamboo in taxa from Group III was superior, and the diameter at breast height was similar to that of taxa from Group I, but the length of the node at breast height was shorter. Increasing the specific leaf area and keeping the nutrients in the body can allow the plant to better adapt to the rich environment [34]. In the present study, taxa from Group IV had a smaller leaf area and leaf length and a larger specific leaf area, reflecting that taxon IV may have better environmental adaptation.

## 5. Conclusions

The phenotypic traits of China’s moso bamboo germplasm resources have a certain degree of variation and differentiation, and in general, the diversity of phenotypic traits is low. Seven key phenotypic indices, namely diameter at breast height (DBH), leaf area (LA), leaf weight (LW), branch-to-leaf ratio (BLr), leaf water content (LWC), wall-to-cavity ratio (WCR) and node length (NL), were selected to evaluate the moso bamboo germplasm resources. Based on the 28 phenotypic traits, China’s moso bamboo germplasm was divided into four groups, each with its characteristics, and important candidate germplasm could be screened out based on the characteristics of each group and the D-value, which could provide a reference for the development and utilization of moso bamboo resources.

## Figures and Tables

**Figure 1 plants-13-01625-f001:**
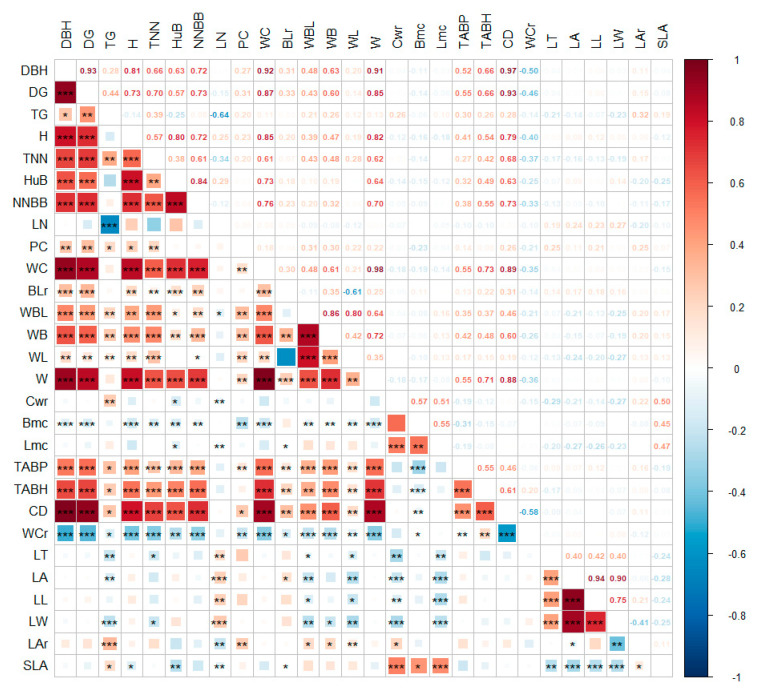
Correlation between phenotypic traits of moso bamboo. *: *p* < 0.05; **: *p* < 0.01; ***: *p* < 0.001.

**Figure 2 plants-13-01625-f002:**
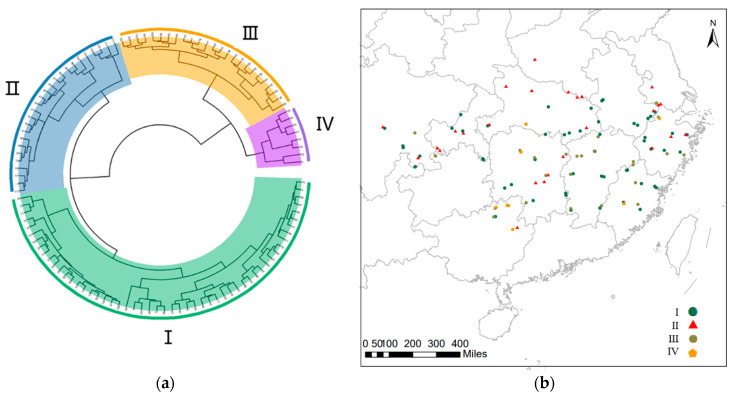
113 moso bamboo germplasm resource clusters: (**a**) cluster map of 113 moso bamboo germplasm resources; (**b**) distribution of different taxa of moso bamboo in China.

**Figure 3 plants-13-01625-f003:**
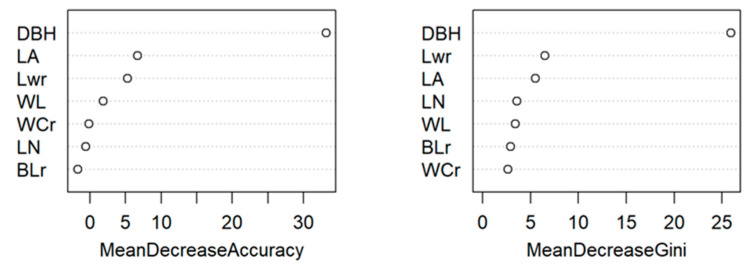
Degree of importance of phenotypic traits in moso bamboo.

**Table 1 plants-13-01625-t001:** Diversity analysis of phenotypic traits of moso bamboo.

Trait	Minimum Value	Maximum Value	Mean Value	Standard Deviation	Coefficient of Variation/%
DBH/cm	6.45	12.73	10.28	1.23	11.97
DG/cm	7.13	15.27	11.82	1.52	12.84
TG/cm * m^−1^	0.52	1.14	0.75	0.09	11.84
H/m	12.08	19.67	15.7	1.54	9.84
TNN/node	50	72	63.27	4.75	7.51
HuB/m	4.36	10.87	7.21	1.34	18.56
NNuB/node	18	35	27.22	3.17	11.65
LN/cm	20.83	30	24.35	1.83	7.53
PC/m	1.58	3.02	2.27	0.25	11.06
WC/kg	9.73	44.45	26.44	6.91	26.14
BLr/g * g^−1^	0.72	3.56	1.72	0.57	33.33
WBL/kg	2.36	11.03	6.49	1.75	26.9
WB/kg	1.65	6.9	3.85	1.14	29.65
WL/kg	0.53	4.99	2.64	0.9	33.96
W/kg	13.62	52.82	32.91	7.97	24.23
Cwr/g * g^−1^	0.51	1.63	0.88	0.19	21.84
Bmc/g * g^−1^	0.42	1.66	0.61	0.13	21.15
Lmc/g * g^−1^	0.49	4.54	1.1	0.4	36.57
TABP/mm	9.85	21.71	16.54	1.99	12.03
TABH/mm	6.66	16.04	10.22	1.3	12.68
CD/mm	49.21	104.32	80.48	10.38	12.89
WCr/mm * mm^−1^	0.2	0.39	0.26	0.03	10.78
LT/mm	0.09	0.16	0.13	0.01	11.13
LA/cm^2^	6.8	15.14	10.45	1.49	14.24
LL/cm	7.29	12.08	10.05	0.75	7.49
LW/cm	1.21	1.81	1.48	0.12	7.81
LAr/cm * cm^−1^	5.24	7.91	6.82	0.4	5.87
SLA/cm^2^ * g^−1^	129.26	271.07	182.59	26.83	14.69
Mean					16.65

Note: DBH. Diameter at breast height; DG. Diameter at ground; TG. Taper grade; H. Height; TNN. Total number of nodes; HuB. Height under branch; NNuB. Number of nodes under branch; LN. Length of node at breast diameter; PC. Plant crown; WC. Weight of culms WN; BLr. Branch-to-leaf ratio; WBL. Weight of branches and leaves; WB. Weight of branches; WL. Weight leaves; W. Total weight; Cwr. Culm moisture content; Lmc. Leaf moisture content; Bmc. Branch moisture content; TABP. Thickness at base of pole; TABH. Thickness at breast height; CD. Cavity diameter; WCr. Wall-to-cavity ratio; LT. Leaf thickness; LA. Leaf area; LL. Leaf length; LW. Leaf width; LAr. Leaf aspect ratio; SLA. Specific leaf area.

**Table 2 plants-13-01625-t002:** Principal component analysis of 28 phenotypic traits.

	Comp. 1	Comp. 2	Comp. 3	Comp. 4	Comp. 5	Comp. 6	Comp. 7	Comp. 8
DBH	0.105	0.006	−0.022	0.053	−0.008	−0.003	−0.036	0.009
DG	0.103	−0.001	−0.012	0.093	−0.077	0.001	0	−0.045
TG	0.038	−0.102	0.085	0.235	−0.247	0.076	0.071	−0.215
H	0.088	0.095	−0.066	−0.063	0.066	0.045	−0.154	0.124
TNN	0.083	−0.059	−0.008	0.02	−0.055	0.025	−0.03	−0.21
HuB	0.069	0.111	−0.209	−0.116	0.022	−0.011	−0.169	0.014
NNBB	0.083	0.033	−0.189	−0.023	−0.074	−0.042	−0.039	−0.174
LN	−0.014	0.15	−0.037	−0.127	0.244	−0.027	−0.186	0.371
PC	0.031	0.038	0.188	0.091	0.079	0.117	0.086	0.292
WC	0.102	0.057	−0.053	−0.034	0.007	−0.014	0.029	0.019
BLr	0.012	0.072	−0.158	0.3	0.016	−0.105	0.397	0.279
WBL	0.071	−0.055	0.215	−0.107	0.204	0.044	0.16	−0.053
WB	0.074	−0.023	0.143	0.017	0.176	0.007	0.343	0.108
WL	0.043	−0.077	0.237	−0.23	0.172	0.077	−0.125	−0.24
W	0.104	0.037	0.001	−0.054	0.053	−0.003	0.059	0.004
Cwr	0	−0.108	−0.084	0.197	0.105	0.284	−0.349	0.113
Bmc	−0.016	−0.056	−0.132	0.122	0.22	0.175	0.024	−0.169
Lmc	0.002	−0.027	−0.109	0.157	0.269	0.121	−0.008	−0.228
TABP	0.063	0.028	0.103	0.065	−0.084	−0.176	0.057	−0.053
TABH	0.076	−0.011	−0.074	−0.126	−0.152	0.375	0.152	0.168
CD	0.103	0.001	−0.035	0.067	0.01	−0.016	−0.089	0.005
WCr	−0.038	−0.012	−0.048	−0.223	−0.184	0.452	0.287	0.174
LT	−0.014	0.119	0.12	0.103	0.099	−0.177	0.143	0.042
LA	−0.021	0.21	0.092	0.097	−0.016	0.242	−0.036	−0.227
LL	−0.008	0.191	0.161	0.153	−0.06	0.245	−0.136	−0.076
LW	−0.029	0.209	0.004	0.02	0.046	0.165	0.114	−0.333
LAr	0.026	−0.032	0.201	0.174	−0.136	0.081	−0.345	0.366
SLA	−0.007	−0.122	−0.062	0.118	0.255	0.214	0.095	0.071
Eigenvalue	9.241	3.985	2.335	1.967	1.83	1.29	1.185	1.026
Contribution rate	33.002	14.231	8.338	7.026	6.536	4.608	4.232	3.663
Cumulative contribution rate	33.002	47.234	55.572	62.598	69.134	73.742	77.974	81.637

**Table 3 plants-13-01625-t003:** Comprehensive score and ranking of 113 moso bamboo germplasm.

Number	D	Rank	Number	D	Rank
Huangshan 1	0.55	62	Jiujiang 1	0.445	100
Huangshan 2	0.556	61	Jiujiang 2	0.692	9
Guangde 1	0.695	8	Yifeng 1	0.557	59
Guangde 2	0.466	97	Yifeng 2	0.5	82
Ningguo 1	0.599	44	Anfu 1	0.593	48
Ningguo 2	0.584	51	Anfu 2	0.638	25
Huoshan 1	0.785	2	Shangrao 1	0.51	79
Huoshan 2	0.634	28	Shangrao 2	0.57	55
Dehua 1	0.644	23	Yihuang 1	0.531	69
Dehua 2	0.689	10	Yihuang 2	0.515	77
Yongan 1	0.427	105	Ruijin 1	0.504	81
Yongan 2	0.612	39	Ruijin 2	0.427	105
Wuyi 1	0.803	1	Chongyi 1	0.594	47
Wuyi 2	0.703	5	Chongyi 2	0.569	56
Jianou 1	0.716	3	Fenghua 1	0.667	15
Jianou 2	0.668	14	Fenghua 2	0.493	84
Jiaocheng 1	0.664	17	Huangyan 1	0.686	12
Jiaocheng 2	0.647	21	Huangyan 2	0.499	83
Nanzhao 1	0.405	108	Jinyun 1	0.648	20
Shihe 1	0.493	84	Jinyun 2	0.621	34
Xinxian 1	0.614	38	Longyou 1	0.522	73
Xinxian 2	0.487	90	Longyou 2	0.517	74
Yiliang 1	0.533	68	Anji 1	0.583	52
Yiliang 2	0.684	13	Anji 2	0.467	96
Changning 1	0.535	67	Zhuji 1	0.623	33
Changning 2	0.638	25	Zhuji 2	0.478	91
Muchuan 1	0.631	29	Chun’an 1	0.616	36
Muchuan 2	0.617	35	Chun’an 2	0.563	58
Tianquan 1	0.624	32	Jurong 1	0.445	100
Tianquan 2	0.491	87	Yixing 1	0.426	107
Zizhong 1	0.643	24	Yixing 2	0.489	88
Pingle 1	0.375	111	Liyang 1	0.583	52
Pingle 2	0.317	113	Chibi 1	0.548	63
Xing’an 1	0.517	74	Chibi 2	0.599	44
Xing’an 2	0.46	98	Yangxin 1	0.405	108
Sanjiang 1	0.54	66	Yangxin 2	0.591	50
Sanjiang 2	0.528	70	Huangmei 1	0.701	6
Rong’an 1	0.492	86	Lutian 1	0.665	16
Rong’an 2	0.647	21	Jingshan 1	0.472	94
Pingjiang 1	0.608	40	Shishou 1	0.489	88
Pingjiang 2	0.441	103	Enshi 1	0.699	7
Taojiang 1	0.629	31	Enshi 2	0.445	100
Taojiang 2	0.544	65	Yidu 1	0.596	46
Taoyuan 1	0.548	63	Nanzhang 1	0.382	110
Taoyuan 2	0.607	41	Zhushan 1	0.374	112
Xiangtan 1	0.516	76	Changshou 1	0.476	92
Xiangtan 2	0.474	93	Changshou 2	0.526	71
Hengyang 1	0.526	71	Liangping 1	0.659	18
Hengyang 2	0.506	80	Fengdu 1	0.429	104
Suining 1	0.638	25	Fengdu 2	0.708	4
Suining 2	0.605	42	Xiushan 1	0.631	29
Shuangpai 1	0.557	59	Xiushan 2	0.689	10
Shuangpai 2	0.592	49	Jiangjin 1	0.514	78
Yanling 1	0.576	54	Jiangjin 2	0.468	95
Yanling 2	0.565	57	Chishui 1	0.615	37
Wanli 1	0.451	99	Chishui 2	0.6	43
Wanli 2	0.649	19			

**Table 4 plants-13-01625-t004:** Correlation coefficients between composite scores (D-value) and phenotypic traits.

Trait	D	Trait	D
DBH	0.851 ***	W	0.863 ***
DG	0.815 ***	Cwr	−0.105
TG	0.184 *	Bmc	−0.144 **
H	0.767 ***	Lmc	−0.062
TNN	0.538 ***	TABP	0.548 ***
HuB	0.500 ***	TABH	0.572 ***
NNBB	0.539 ***	CD	0.816 ***
LN	0.107	WCr	−0.395 ***
PC	0.501 **	LT	0.219
WC	0.826 ***	LA	0.250
BLr	0.372	LL	0.333
WBL	0.611 ***	LW	0.171
WB	0.748 ***	LAr	0.217 *
WL	0.262 **	SLA	−0.032 *

Note: *: *p* < 0.05; **: *p* < 0.01; ***: *p* < 0.001.

**Table 5 plants-13-01625-t005:** Comparison of phenotypic traits of different groups of moso bamboo germplasm resources.

Trait	Items	Group			
		I	II	III	IV
DBH	M ± SD	11.05 ± 0.8 a	8.88 ± 1.03 d	10.28 ± 0.55 b	9.7 ± 1.48 c
DG	M ± SD	12.77 ± 1.03 a	10.1 ± 1.25 d	11.79 ± 0.62 b	11.03 ± 1.81 c
TG	M ± SD	0.76 ± 0.07 a	0.7 ± 0.08 b	0.78 ± 0.07 a	0.76 ± 0.17 a
H	M ± SD	16.66 ± 1.39 a	14.37 ± 0.91 b	15.3 ± 0.91 bc	14.81 ± 1.62 c
TNN	M ± SD	65.37 ± 3.55 a	58.92 ± 5.01 b	64.12 ± 3 a	60.63 ± 5.55 b
HuB	M ± SD	7.98 ± 1.29 a	6.34 ± 0.9 b	6.61 ± 1.01 b	6.68 ± 0.88 b
NNBB	M ± SD	29.17 ± 2.59 a	24.31 ± 2.69 c	26.4 ± 2.18 b	26.13 ± 1.81 b
LN	M ± SD	24.32 ± 1.79 ab	25.05 ± 2.17 a	23.72 ± 1.36 b	24.28 ± 1.82 ab
PC	M ± SD	2.31 ± 0.27 a	2.2 ± 0.2 a	2.25 ± 0.27 a	2.22 ± 0.21 a
WC	M ± SD	31.07 ± 5.47 a	19.56 ± 4.82 c	24.91 ± 3.59 b	22.24 ± 6.4 bc
BLr	M ± SD	1.81 ± 0.51 a	1.51 ± 0.57 a	1.7 ± 0.65 a	1.86 ± 0.65 a
WBL	M ± SD	6.84 ± 1.52 ab	5.27 ± 1.45 c	7.2 ± 1.8 a	5.93 ± 1.99 bc
WB	M ± SD	4.19 ± 1.04 a	2.93 ± 0.67 b	4.19 ± 1.24 a	3.56 ± 1.14 ab
WL	M ± SD	2.66 ± 0.77 ab	2.34 ± 0.92 b	3.01 ± 1 a	2.37 ± 0.96 b
W	M ± SD	37.88 ± 6.28 a	24.83 ± 5.85 c	32.11 ± 4.54 b	28.17 ± 8.09 bc
Cwr	M ± SD	0.83 ± 0.19 a	0.88 ± 0.21 a	0.96 ± 0.15 a	0.93 ± 0.22 a
Bmc	M ± SD	0.58 ± 0.07 c	0.6 ± 0.08 bc	0.67 ± 0.21 ab	0.71 ± 0.16 a
Lmc	M ± SD	1.08 ± 0.54 a	1.06 ± 0.19 a	1.13 ± 0.14 a	1.31 ± 0.32 a
TABP	M ± SD	17.37 ± 1.86 a	15.31 ± 1.63 b	16.21 ± 1.79 ab	15.88 ± 2.18 b
TABH	M ± SD	86.69 ± 7.36 a	69.17 ± 9.08 c	80.23 ± 4.85 ab	76.18 ± 11.98 bc
CD	M ± SD	10.77 ± 1.08 a	9.14 ± 1.2 c	10.34 ± 1.07 b	9.72 ± 1.42 b
WCr	M ± SD	0.25 ± 0.03 a	0.27 ± 0.03 a	0.26 ± 0.03 a	0.26 ± 0.02 a
LT	M ± SD	0.13 ± 0.01 a	0.13 ± 0.02 a	0.12 ± 0.01 a	0.13 ± 0.01 a
LA	M ± SD	10.66 ± 1.18 a	10.61 ± 2.04 a	10.26 ± 1.13 a	9.15 ± 1.77 b
LL	M ± SD	10.2 ± 0.55 a	10 ± 0.97 a	10.06 ± 0.55 a	9.07 ± 1.06 b
LW	M ± SD	1.49 ± 0.1 a	1.5 ± 0.16 a	1.44 ± 0.09 a	1.44 ± 0.14 a
LAr	M ± SD	6.87 ± 0.31 ab	6.69 ± 0.33 b	7.01 ± 0.29 a	6.35 ± 0.83 c
SLA	M ± SD	166.91 ± 16.42 c	176.21 ± 13.32 c	202.19 ± 9.89 b	247.93 ± 14.37 a
D	M	0.62	0.47	0.56	0.5

Note: Different letters in indicate significant difference at *p* < 0.05.

**Table 6 plants-13-01625-t006:** Classification prediction results of moso bamboo germplasm in different taxa.

Sample Type	Training Set				Test Set			
	I	II	III	IV	I	II	III	IV
Number of samples	41	18	13	7	12	11	10	1
Accurate prediction number	34	14	7	3	9	6	9	0
Prediction accuracy/%	82.93%	77.78%	53.85%	42.86%	75.00%	54.55%	90.00%	0.00%
Average accuracy/%	73.42%				70.59%			

## Data Availability

The original contributions presented in the study are included in the article, further inquiries can be directed to the corresponding author.

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
