# Peer review of "The Phenotypic Variation in Moso Bamboo and the Selection of Key Traits"

_plants, 2024, doi:10.3390/plants13121625_

Round 1

Reviewer 1 Report

Comments and Suggestions for Authors

In this work, the authors investigated investigate 28 phenotypic traits of Moso bamboo across the entire distribution area of China using the grid method. The results could provide a reference for the development and utilization of Moso bamboo resources. The topic of the manuscript is relevant with the aims of Plants. The manuscript is original. However, there are still some issues and problems need to be addressed before the MS can be accepted for publication in Plants.

Specific comments:

Line 13: ‘the diversity of phenotypic traits of moso bamboo is low’, but, there is no mention of diversity calculations throughout the manuscript. And in line 136 it says high diversity, is there a contradiction here? Please check the results and provide calculations.

Line 17: The text mentions the use of random forests for result validation analysis, but he entire text does not mention the calculation parameters and results of random forests. Please provide the relevant information.

Line 19-20: The text evaluates germplasm resources by D values, however, samples were taken from many different regions. Is it scientific to evaluate phenotypic traits without considering environmental factors? Please provide relevant literature as a basis.

Line 62:  “150km*150km” should be added space.

Line 157:  change “Fig. 1” to “ Figure 1”.

Line 204: Seven ‘key’ phenotypic traits (breast diameter, leaf area, leaf weight, etc.,) were selected from 28 phenotypic traits. May I ask what are the criteria and calculation process for this selection?

Line 300: The proper way of citing references should be noted.

References: please check the format, and delete [J], [S].

The language expression should be improved.

Comments on the Quality of English Language

 Minor editing of English language required.

Author Response

Thank you very much for taking the time to review this manuscript. Please find the detailed responses below and the corresponding revisions/corrections highlighted/in track changes in the re-submitted files. Should you have any further questions or require additional clarification, please do not hesitate to reach out to us.

Reviewer 2 Report

Comments and Suggestions for Authors

The authors performed interesting research on morphological parameters of bamboo trees from several locations in China. My comments are given directly in the document. I hope that the authors will find them well-intended and useful.

However, I have to clearly state here that I do not feel competent for judging some statistical procedures applied in this manuscript. I am not familiar with the calculation and discussion of comprehensive index score (D) and affiliation function method for evaluation of germplasm resources. Therefore, I kindly asked the editor to find another reviewer who is competent in this field, in order not to miss some important errors in interpretation of the results. Significant part of the discussion is based on the results of these calculations, it is very important.

Author Response

  1. “essential features indicative of moso bamboo”—— what is the meaning of this? indicative for what?

I changed essential features indicative of moso bamboo to “essential characteristics of moso bamboo”(line 10). I'd like to go over a few key traits that can be used to evaluate moso bamboo.

  1. 87% to 36.57%, the diversity of phenotypic traits of moso bamboo is low. ——
    1. line 130 says that bamboo was "rich in variation".
    2. This statement is too general, 36% is not low.

Thank you for your careful attention. I have made the modifications. (line 12).

36% was not low. However, the mean coefficient of variation of phenotypic traits of moso bamboo was 16.65%, which were lower than those of Dendrocalamus lactiferous Munro (30.84%), Salix psammophila (22.53%), and Ziziphus jujuba var. spinosa (Bunge) Hu ex H.F.Chow. (19. 80%) (Hao et al., 2017; Li et al., 2023; Qu et al., 2024), indicating that moso bamboo germplasm is less variable than other species.

  1. identify seven main phenotypic trait indicators——

Not all traits were phenotypic (e.g. moisture content)

Moisture content is often considered a physiological trait that reflects the state and regulatory mechanisms of water in an organism and is often used to assess plant adaptation to drought or water stress.

  1.  

Your attention to detail is greatly appreciated. We have carefully reviewed your suggestions and have corrected the errors accordingly.(line33, line35)

  1. There are significant differences in the growth and phenotypic traits of moso bamboo populations——Which traits? Which differences? Say precisely.

These research results provide a good basis——The results suggest that the phenotypic traits of moso bamboo exhibit rich diversity, with coefficients of variation ranging from 5.87% to 36.57%.

which revealed the phenotypic diversity of moso bamboo germplasm in China, explored the relationship between the phenotypic traits of moso bamboo, and screened the key phenotypic traits,——Please write this sentence in the form of Aim of the research. Here the authors pointed already to the results. Say clearly what was the aim of this research - The aim was to reveal the phenotypic diversity of moso bamboo and explore the realtionship between........in order to provide…

Thank you. The introductory section has been revised based on your suggestions. (line 37-67)

  1. wall thickness

Your attention to detail is greatly appreciated. I have changed " wall thickness " to " bamboo wall thickness ". (line 95)

  1. killed

Your attention to detail is greatly appreciated. I have changed "killed" to "dried". (line 103)

  1. Dali et al, 104 2022); random forest modeling of phenotypic trait data was achieved using the R extension package random Forest (Meng et al, 2022);——I am not sure if these references are adequate to cite here. These are just papers in which the same methods were applied, not methodological papers. Please find original references which explain these statistical analyses.

 I have made the modification. (line 113-115)

  1. The coefficients of variation of the phenotypic traits of moso bamboo ranged from 5.87% to 36.57% (Table 1), indicating that the numerical traits of moso bamboo were rich 130 in variation——rich or low

I have made the modifications. (line 12).

  1. Figure——I find it unnecessary to give the same results twice in this figure. Please leave just one half of the figure (just squares-which I would personally choose, or just circles), and delete the other.

I have made the modification. (line 166-183)

  1. 3 Same text given twice!!!!

I have made the modification. (line 203-206)

  1. I do not think that so many (8) components are needed. 4 would be just enough. The text is hard to follow and not informative after 4th component.

The experiment selected 28 traits and retained principal components with eigenvalues greater than one, achieving a cumulative contribution rate of 81.63%. If only the first four principal components are chosen, many important leaf traits will be lost.

  1. correlation coefficients between breast diameter 313ground diameter——this is expectable correlation, would not mention at all

Based on the results of the relevant analysis, it can be concluded that there is a significant correlation between breast diameter and ground diameter, with a correlation coefficient greater than 0.9. For detailed information, please refer to section 3.2 (line 166).

  1. studying their relationship and 319converting a large number of correlated traits into a few uncorrelated traits can more 320clearly show the role of each phenotype in the——I do not understand what the authors wanted to say here.

I have made the modification. (line 347-352)

  1. (Jan, 2020). —— 2019

Your attention to detail is greatly appreciated.(line 359)

  • Below is a detailed description of our criteria and calculations for selecting these phenotypic traits:

1 Correlation analysis: We first performed a correlation analysis on the 28 phenotypic traits to determine the degree of correlation between them.

2 Principal component analysis (PCA): We performed principal component analysis on all phenotypic traits to determine which traits best represented the variation in the overall data. We selected the representative traits in each principal component based on the contribution of each trait in each principal component, and initially selected seven key phenotypic traits.

3 Random forest: The random forest discriminant model was applied to validate the application of the seven key phenotypic indicators. It can better predict the classification discrimination of four taxa.

  • I am adding a special section 3.6 to the manuscript that details the computational parameters and results of the random forest model. See lines 289-310 for details.

Round 2

Reviewer 1 Report

Comments and Suggestions for Authors

The current version of the manuscript can be accepted for publication.